# BatchQuant: Quantized-for-all Architecture Search with Robust Quantizer

**Haoping Bai**[*]   **Meng Cao**   **Ping Huang**   **Jiulong Shan**

Apple

{haoping_bai, mengcao, huang_ping, jiulong_shan}@apple.com

## Abstract

As the applications of deep learning models on edge devices increase at an accelerating pace, fast adaptation to various scenarios with varying resource constraints has become a crucial aspect of model deployment. As a result, model optimization strategies with adaptive configuration are becoming increasingly popular. While single-shot quantized neural architecture search enjoys flexibility in both model architecture and quantization policy, the combined search space comes with many challenges, including instability when training the weight-sharing supernet and difficulty in navigating the exponentially growing search space. Existing methods tend to either limit the architecture search space to a small set of options or limit the quantization policy search space to fixed precision policies. To this end, we propose BatchQuant, a robust quantizer formulation that allows fast and stable training of a compact, single-shot, mixed-precision, weight-sharing supernet. We employ BatchQuant to train a compact supernet (offering over $10^{76}$ quantized subnets) within substantially fewer GPU hours than previous methods. Our approach, Quantized-for-all (QFA), is the first to seamlessly extend one-shot weight-sharing NAS supernet to support subnets with arbitrary ultra-low bitwidth *mixed-precision* quantization policies *without retraining*. QFA opens up new possibilities in joint hardware-aware neural architecture search and quantization. We demonstrate the effectiveness of our method on ImageNet and achieve SOTA Top-1 accuracy under a low complexity constraint ($< 20$ MFLOPs). The code and models will be made publicly available at https://github.com/bhpfelix/QFA.

## 1   Introduction

In order to deploy deep learning models on resource-constrained edge devices, careful model optimization, including pruning and quantization is required. While existing works have demonstrated the effectiveness of model optimization techniques in speeding up model inference [1, 2, 3, 4], model optimization increases human labor by introducing extra hyperparameters. Consequently, automated methods such as neural architecture search (NAS) [5, 6, 7, 8, 9, 10, 11, 12, 13, 14, 15, 16] and automated quantization policy search [17, 18, 19] have emerged to alleviate the human bandwidth required for obtaining compact models with good performance.

In this paper, we focus on finding the best of both worlds—the best combination of architecture and mixed-precision quantization policy. However, combining two complex search spaces is inherently challenging, not to mention that quantization usually requires a lengthy quantization-aware training (QAT) procedure to recover performance. Thus, previous methods tend to employ proxies to estimate the performance of an architecture and quantization policy combination. For example, APQ [20] performs QAT on each of 5000 architecture and quantization policy combinations for 0.2 GPU hours

---

[*]Corresponding author: Haoping Bai haoping_bai@apple.com

35th Conference on Neural Information Processing Systems (NeurIPS 2021).

Table 1: Comparisons of quantized architecture search approaches: DNAS [19] SPOS [12], HAQ [17], APQ [20], OQA [23] and BQ (Ours). "Single-shot mixed-precision QAT" means the supernet is directly trained with QAT on arbitrary mixed-precision quantization policies. "No training during search" means there is no need for re-training the sampled network candidate during the search phase, and this is accomplished by single-shot supernet training similar to [22]. "No evaluation during search" means that we do not have to evaluate sampled network candidates on the validation dataset during the search phase, and this is achieved by training an accuracy predictor similar to [22]. "No retraining/finetuning" means that we do not have to finetune any searched architectures as weights inherited from supernet already allow inference for the given architecture under the specified quantization policy. "Weight-sharing" means that quantization policies share the same underlying full-precision weights so that we can obtain mixed-precision weights by fusing the corresponding quantizers with the same set of full precision weights. "Compact MobileNet search space" means that supernet is based on the Mobilenet search space, which offers better accuracy v.s. complexity trade-off but is more sensitive to quantization than heavy search space such as ResNet.

|  | DNAS | SPOS | HAQ | APQ | OQA | **QFA** |
|---|---|---|---|---|---|---|
| Single-shot mixed-precision QAT |  | ✓ |  |  |  | ✓ |
| No training during search |  | ✓ |  |  | ✓ | ✓ |
| No evaluation during search |  |  |  | ✓ | ✓ | ✓ |
| No retraining / finetuning |  |  |  |  | ✓ | ✓ |
| Mixed-precision quantization | ✓ | ✓ | ✓ | ✓ |  | ✓ |
| Weight-sharing |  | ✓ |  |  |  | ✓ |
| Compact MobileNet search space |  |  | ✓ | ✓ | ✓ | ✓ |

and uses the sampled combinations to train an accuracy predictor. BATS [21] searches for cell structures that are repeatedly stacked to form the target architecture. Proxy-based methods necessitate both careful treatments to ensure reliable ranking of quantized architectures and a time-consuming retraining procedure when the target quantized architecture is identified, rendering proxy-based approaches impractical for frequently changing deployment scenarios.

To avoid the lengthy retraining process, NAS methods that train a single-shot weight-sharing supernet [12, 22] are ideal. However, many previous works have shown evidence that QAT of mixed-precision supernets can easily become highly unstable [23, 21]. As a result, existing single-shot quantized architecture search methods usually limit the size of the combined search space. For example, SPOS [12] sacrifices architecture search space size to only allow channel search and requires retraining to recover performance. OQA [23] limits its quantization policy search space to fixed-precision quantization policies and trains a separate set of weights for each bitwidth.

To successfully train the mixed-precision supernet, we propose BatchQuant (BQ). Analogous to batch normalization, BQ leverages batch statistics to adapt to the shifting activation distribution as a result of quantized subnet selection, offering better robustness to outliers than quantizer with vanilla running min/max based scale estimation, and better flexibility than a learnable quantizer that only learns a fixed set of parameters. Without limiting the architecture search space, our joint architecture and quantization policy search space contains over $10^{76}$ possible quantized subnets, providing much more flexibility than previous search spaces (e.g. The OQA search space has $10^{20}$ possible quantized subnets). While our approach and OQA both follow the supernet training strategy introduced in [22], our weight-sharing supernet takes only 190 epochs to train despite the complex search space, significantly less than the 495 epochs required by OQA. We further leverage the NSGA-II algorithm to produce a Pareto set of quantized architectures that densely covers varying complexity constraints, eliminating the marginal cost of adapting architecture to new deployment scenarios.

The contributions of the paper are

- To the best of our knowledge, we present the first result to train one-shot weight-sharing supernet to support subnets with arbitrary mixed-precision quantization policy *without retraining*.

- We propose BatchQuant, an activation quantizer formulation for stable mixed-precision supernet training. The general formulation of BatchQuant allows easy adaptation of new scale estimators.

- Compared with existing methods, our method, QFA, takes a shorter time to train a significantly more complex supernet with over $10^{76}$ possible quantized subnets and discovers quantized subnets at SOTA efficiency with no marginal search cost for new deployment scenarios.

Table 2: Design runtime comparison with state-of-the-art quantized architecture search methods. Here we use $T(\cdot)$ to denote the runtime of an algorithm. Note we separate the accuracy predictor training from the search procedure because it is a one-time cost amortized across deployment scenarios. We define marginal cost as the cost for searching in a new deployment scenario, and we use $N$ to denote the number of deployment scenarios. Our method eliminates the marginal search cost completely.

| Method | Design Runtime |
|--------|----------------|
| SPOS | T(Supernet Training) + T(Search + Finetune) × N |
| APQ | T(Supernet Training + Accuracy Predictor Training) + T(Search + Finetune) × N |
| OQA | T(Supernet Training + Accuracy Predictor Training) + T(Search + Finetune) × N |
| Ours | T(Supernet Training + Accuracy Predictor Training + Search) |

## 2 Related Work

### 2.1 Mixed Precision Quantization

Different layers of a network have different redundancy and representation power, each layer will react to quantization differently and may achieve varying levels of efficiency gain on hardware. In fact, many fixed-bitwidth quantization methods are inherently mixed-precision by making design decisions to leave the first convolution layer, batch-norm layers [24], squeeze and excitation layers [25], and the last fully connected layer at full precision / int8 precision [3, 2, 19, 12, 23]. As a result, mixed-precision quantization methods [26, 18, 27, 17] emerge in the place of fixed precision quantization and are receiving increasing attention from hardware manufacturers [17]. In contrast to mixed-precision quantization methods that optimize for a single quantization policy on a single architecture, QFA allows the discovery of the most suitable quantization policy for arbitrary subnet architectures.

**Adaptive Quantization** is also closely related to our work. Methods include Adabits [28], Any-Precision DNN [29], Gradient $\ell_1$ regularization or quantization robustness [30], and KURE [31] train model that can adaptively switch to different bitwidth configuration during inference. Most methods [28, 29, 30] only address the case of fixed precision quantization where all weights share the same bitwidth and activations share another. Our quantization policy search space offers much more flexibility in terms of layerwise mixed-precision quantization.

### 2.2 Joint Mixed-Precision Quantization and Neural Architecture Search

There exist many exciting works in the intersection of mixed precision quantization and NAS. JASQ [32] employs population-based training and evolutionary search to produce a quantized architecture under a combined accuracy and model size objective. DNAS [19] is a differentiable NAS method that optimizes for a weighted combination of accuracy and resource constraints. SPOS [12] trains a quantized one-shot supernet to search for bit-width and network channels for heavy ResNet search space. APQ [20] builds upon a full precision one-shot NAS supernet and trains 5000 quantized subnets to build a proxy accuracy predictor. The best quantized architecture proposed by the predictor is then retrained. Most existing methods fall into two categories. The first optimizes for a single weighted objective of accuracy and complexity and produces only a single or a few quantized architectures [32, 19], which is difficult to scale to multiple deployment scenarios. The second category is capable of estimating the performance for many quantized architectures by using proxies such as weight-sharing supernet and partially trained models [12, 20]. Due to the accuracy degradation in weight-sharing supernet in SPOS and the use of proxy in APQ, both methods require retraining when the best quantized architecture is discovered. On the contrary, our approach does not require retraining. Training the weight-sharing supernet with BQ reduces accuracy degradation, allowing quantized subnets to reach competitive performance without retraining. As a result, QFA enjoys no marginal cost for deploying quantized subnet to new scenarios. Table 1 details the difference of our approach from other architecture search approaches. Table 2 compares the algorithmic complexity of our method with other mixed-precision quantized architecture search methods.

# 3 Stabilizing Mixed Precision Supernet Training with BatchQuant

We adopt the single-path weight-sharing MobileNetV3 [33] supernet formulated in [22] that supports adaptive input resolution, network kernel size, depth, and width. In addition, we assign simulated quantization operations (quantizers) [34], one for each bitwidth, to each weight and activation tensor within the supernet to support quantization-aware training (QAT). We allow bitwidths $\{2, 3, 4\}$ for both weight quantizers and activation quantizers.

The compact weight-sharing supernet formulation allows us to share a single set of quantizers across all the subnets. However, such compactness also leads to instability. Specifically, supernet training becomes highly unstable when activations from different subnets are quantized by a shared quantizer. In the following sections, we will review the fundamentals of quantization and address why shared activation quantizer can cause unstable supernet training. Then, we introduce BatchQuant as a solution to stabilize supernet training.

## 3.1 Quantization Preliminaries

To help address the difficulty in training the mixed-precision supernet, we start by introducing common notations for quantization. WLOG, we consider the case of uniform affine (asymmetric) quantization. Let $\boldsymbol{x} = \{x_1, \cdots, x_N\}$ be a floating point vector/tensor with range $(x_{min}, x_{max})$ that needs to be quantized to $b$-bitwidth precision. The integer coding $\boldsymbol{x}_q$ will have range $[n, p] = [0, 2^b - 1]$. Then we derive two parameters: Scale ($\Delta$) and Zero-point($z$) which map the floating point values to integers (See [1]). The scale specifies the step size of the quantizer and floating point zero maps to zero-point [34], an integer which ensures that zero is quantized with no error. The procedure is as follows:

$$\Delta = \frac{x_{max} - x_{min}}{p - n}, \quad z = clamp\left(-\left\lfloor \frac{x_{min}}{\Delta} \right\rceil, n, p\right)$$

$$\boldsymbol{x}_q = \left\lfloor clamp\left(\frac{\boldsymbol{x}}{\Delta} + z, n, p\right) \right\rceil \tag{1}$$

$$\hat{\boldsymbol{x}} = (\boldsymbol{x}_q - z)\Delta$$

where $\lfloor \cdot \rceil$ indicates the round function and the $clamp(\cdot)$ function clamps all values to fall between $n$ and $p$. The quantized tensor $\hat{\boldsymbol{x}}$ is then used for efficient computation by matrix multiplication libraries that handle $\Delta$ efficiently (See [35]). Note that during training, the range $(x_{min}, x_{max})$ of activations are not known beforehand. Therefore, standard practice is to keep track of an exponential moving average (EMA) of past extreme values (min&max) to generate the scale estimator $\hat{\Delta}$. As both the calculation of $z$ and $\boldsymbol{x}_q$ relies on a good $\Delta$ estimate, we will later provide insights into why using EMA estimator $\hat{\Delta}$ for activation quantization in the supernet could lead to unstable training.

## 3.2 Weight Quantization with LSQ

Since the value distribution of a weight tensor is relatively stable across updates and is empirically observed to be symmetrical around zero, many methods treat $\Delta$ as a learnable parameter and discard the zero point $z$ [36, 4]. We leverage the symmetric LSQ quantizer [36] to quantize weight tensors in our supernet as follows:

$$\boldsymbol{x}_q = \left\lfloor clamp\left(\frac{\boldsymbol{x}}{\Delta}, n, p\right) \right\rceil$$

$$\hat{\boldsymbol{x}} = \boldsymbol{x}_q\Delta \tag{2}$$

where $[n, p] = [-2^{b-1}, 2^{b-1} - 1]$. Note the rounding operation $\lfloor \cdot \rceil$ has 0 derivative almost everywhere, QAT applies the straight through estimator (STE) [37] to allow gradients to backpropagate through.

## 3.3 Challenges of Activation Quantization in One-shot Supernet

While weight quantization for weight-sharing supernet is similar to that of normal networks, activation quantization requires careful treatment to enable stable supernet training. According to the insights in [31], as well as shown in Equation (1), an unstable $\Delta$ estimate could impact the quantized tensor $\hat{\boldsymbol{x}}$ and negatively impact QAT performance. Given the significant delay in the EMA estimator $\hat{\Delta}$ when

the ranges shift rapidly, practical approaches such as [34] completely disables activation quantization for the first 50 thousand to 2 million steps. However, we now show that one-shot supernet will always have rapidly shifting activation ranges due to subnet sampling.

**EMA Estimator $\hat{\Delta}$ is problematic**. Let $N = B \times C \times H \times W$ denote the number of elements within an activation tensor with batch size $B$, channel number $C$, height $H$, and width $W$. Due to adaptive input resolution and network width (number of channels), the activation map at a given layer will vary in size $(H, W)$ and channel number $C$ depending on the activated subnet. For example, at a batch size of $64$ the size of incoming activations to the second mobile inverted residual block in our search space can vary from $[64, 72, 64, 64]$ to $[64, 144, 112, 112]$.

We now consider the case where elements within $\boldsymbol{x}$ are $i.i.d$ random variables with cumulative distribution $F$, and let $M_N = \max\{x_1, \cdots, x_N\}$ denote the maximum. Then $M_N$ follows the maximum extreme value distribution with the following cumulative distribution function:

$$\mathbb{P}(M_N \leq t) = \mathbb{P}(x_1 \leq t, \cdots, x_N \leq t) = \mathbb{P}(x_1 \leq t) \cdots \mathbb{P}(x_N \leq t) = F(t)^N \qquad (3)$$

Following the assumptions in [38], we model activations within neural networks as tensor-valued Laplace random variables. WLOG, when considering the maximum value distribution, we can model each element as an exponential random variable, because the Laplace distribution can be thought of as two exponential distributions spliced together back-to-back. With $x_i \sim \text{Exp}(\lambda)$, $i \in [1, \cdots, N]$,

$$\mathbb{E}[M_N] = \int_0^\infty \left(1 - F(t)^N\right) dt = \int_0^\infty \left(1 - (1 - e^{-\lambda t})^N\right) dt, \quad \left(\text{let } u = 1 - e^{-\lambda t}\right)$$

$$= \int_0^1 \frac{1}{\lambda} \frac{1 - u^N}{1 - u} du = \frac{1}{\lambda} \sum_{k=1}^N \frac{1}{k} \sim \frac{1}{\lambda} log(N) \qquad (4)$$

We can see that $\mathbb{E}[M_N]$ slowly diverges to $\infty$ as $N$ increases. For a symmetric activation distribution where the minimum and maximum value distributions are symmetric about 0, we can consider $\mathbb{E}[\hat{\Delta}] = 2\mathbb{E}[M_N] \propto log(N)$ for the EMA estimator. As a result, the EMA estimator $\hat{\Delta}$ will never become stable because it is sensitive to the activation tensor size and could change significantly for each update depending on the sampled subnet. For the example mobile inverted residual block given above, the largest difference in $\mathbb{E}[M_N]$ is proportional to $log(1511424 * \text{batch\_size})$, which can lead to substantial instability in the EMA estima-

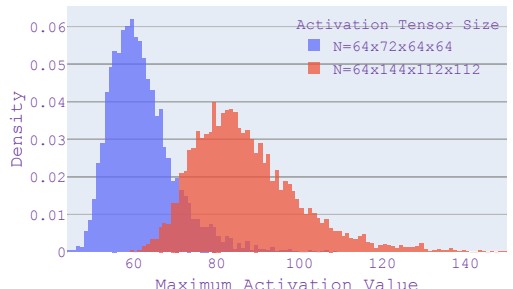

Figure 1: Shifting Max Value Distribution of activation tensors during training.

tion. Furthermore, due to adaptive layer skipping, layers that are only present in deep subnets will activate less frequently and have substantially more lag in EMA update than layers shared among shallow and deep subnets, exacerbating the instability in EMA and the quantized activations. Figure 1 shows the maximum extreme value distribution for the respective activations collected from 1000 batches of training data.

Note such a shifting extreme value distribution problem can also happen to weight tensors in weight-sharing supernets because different subnets can have different weight tensor sizes. Conveniently, our supernet construction ensures that at each layer, the weight tensor for smaller subnets is nested within the weight tensor for the larger subnet. Therefore we simply perform quantization on the entire weight tensor before indexing out the weights for the given subnet to stabilize weight quantization.

**Learnable scale does not help**. Methods that learn a fixed $\Delta$ is also suboptimal in the case of supernet training. According to [38] the optimal quantization level depends on the variance in activation distribution (*e.g.* Laplace parameter $b$ for Laplace distribution or the variance $\sigma^2$ of normal distribution). However, as observed in [29], quantizing weights and input to varying precision leads to activations with different mean and variance. While we can easily resolve this issue in fixed-precision quantization by learning a different set of $\Delta$ for each bitwidth setting, it is impossible to assign a set of $\Delta$ parameters for each mixed-precision quantization policy on each subnet.

Empirically, during the training of our compact MobileNet search space, EMA estimator $\hat{\Delta}$ leads to gradient explosion, and a fixed set of learnable $\Delta$ leads to diverging training loss as well.

## 3.4 Batch Quantization for Robust Activation Quantization

To alleviate the unstable scale estimation issue, we propose the following general definition of batch quantizer (BatchQuant)

$$\hat{\Delta} = \frac{\hat{x}_{max} - \hat{x}_{min}}{p - n}, \quad \hat{z} = clamp\left(-\left\lfloor \frac{\hat{x}_{min}}{\hat{\Delta}} \right\rceil, n, p\right)$$

$$\boldsymbol{x}_q = \left\lfloor clamp\left(\frac{\boldsymbol{x}}{\hat{\Delta}\gamma} + \hat{z} + \beta, n, p\right) \right\rceil \tag{5}$$

$$\hat{\boldsymbol{x}} = (\boldsymbol{x}_q - \hat{z} - \beta)\hat{\Delta}\gamma.$$

Analogous to batch normalization [24], we leverage batch statistics to help standardize the activation distribution across different sampled subnets. Instead of using EMA, we estimate the extreme values $\hat{x}_{min}$ and $\hat{x}_{max}$ **only** from the current batch to negate the effect of changing extreme value distribution due to subnet sampling. In addition, we learn a multiplicative residual $\gamma$ on $\hat{\Delta}$ and an additive residual $\beta$ on $\hat{z}$ to facilitate learning of the optimal clipping range.

Note, similar to LSQ+ [4], zero in activation may not be exactly quantizable based on such a formulation. However, we share emprical observation as LSQ+, that the learned $\beta$ is often small and aids in empirical performance by reducing the quantization error for H-swish activation [33]. Because we use symmetric quantization for weights, asymmetric quantization of activations has no additional cost during inference as compared to symmetric quantization since the bias term can be precomputed and fused with the bias in the succeeding layer:

$$\hat{w}\hat{x} = (w_q\Delta_w)(x_q - \hat{z} - \beta)\hat{\Delta}_x\gamma = w_q x_q \Delta_w \hat{\Delta}_x \gamma - \overbrace{w_q \Delta_w(\hat{z} + \beta)\hat{\Delta}_x\gamma}^{bias}. \tag{6}$$

**BatchQuant (BQ) is generally applicable**. Note our formulation of BatchQuant is general. We do not explicitly define the batch extreme value estimators $\hat{x}_{min}$ and $\hat{x}_{max}$, because it is straightforward to plug in different definitions as the usage sees fit. We will provide three simple example formulations of $\hat{x}_{min}$ and $\hat{x}_{max}$ below. BatchQuant could potentially be a good drop-in replacement for conventional QAT with EMA scale estimator when adjusting training batch size or performing tasks that have varying activation size such as semantic segmentation. Due to the adaptive formulation, BatchQuant offers a much simpler initialization strategy than the optimization-based strategy in LSQ+ [4]. With a proper choice of the batch extreme value estimator $(\hat{x}_{min}, \hat{x}_{max})$, we can simply initialize the residual terms with $\gamma = 1$ and $\beta = 0$. As training is stabilized from the beginning, BQ can also avoid the delayed activation quantization strategy introduced in [34], which requires choosing a starting schedule for activation quantization.

**Extreme Value Estimators** Due to presence of learnable residuals $\gamma$ and $\beta$, the extreme value estimators $(\hat{x}_{min}, \hat{x}_{max})$ does not need to capture the exact value of the scale as long as the calculated scale $\hat{\Delta}$ captures the range variation in activation. Consider 4D activations with vector index $i = (i_B, i_C, i_H, i_W)$ that indexes batch, channel, height, and width dimension respectively, we present three example definitions to use in our experiment:

$$\hat{x}_{min} = \min_i \boldsymbol{x}_i, \qquad\qquad \hat{x}_{max} = \max_i \boldsymbol{x}_i \tag{7}$$

$$\hat{x}_{min} = \frac{1}{C}\sum_{i_C} \min_{i_B, i_H, i_W} \boldsymbol{x}_i, \qquad\qquad \hat{x}_{max} = \frac{1}{C}\sum_{i_C} \max_{i_B, i_H, i_W} \boldsymbol{x}_i \tag{8}$$

$$\hat{x}_{min} = \mu - 3\sigma, \qquad\qquad \hat{x}_{max} = \mu + 3\sigma \tag{9}$$

where $\mu$ is the tensor-wise mean and $\sigma$ is the tensor-wise variance. Note that Equation 7 provides unbiased estimation of extreme values and is analogous to EMA $\Delta$ estimator but only based on batch statistics. Equation 9 is a biased estimation with the least variance and is equivalent to the scale parameter initialization strategy of LSQ+ [4]. Plugging Equation 9 into Equation 5 is thus analogous of using a learned scale quantizer. Equation 8 is a biased estimation with a bias-variance trade-off between Equation 7 and 9. We show empirically that Equation 8 leads to stable supernet training.

**BatchQuant Calibration during test time** Similar to the treatment of batchnorm in [22], to obtain a set of extreme value estimations $(\hat{x}_{min}, \hat{x}_{max})$ specific to a quantized subnet for inference, we simply perform BatchQuant calibration by forwarding a few batches of data and accumulate a running mean of the estimations.

# 4 Mixed-Precision Quantized Architecture Search

Equipped with BatchQuant for stable mixed-precision supernet training, we now introduce the search procedure of our mixed-precision quantized architecture search method, QFA.

**Supernet Search Space Definition** Following the convention [3, 2, 19, 12, 23] of only searching and quantizing the residual blocks in the Mobilenet V3 search space, we keep the input and weights of the first convolutional layer at full precision and quantize the incoming activation and weights of the last fully-connected layer to 8-bit precision. Specifically, our Mobilenet V3 search space contains 5 stages each with 4 mobile inverted residual blocks. Each stage can optionally skip 1 or 2 of its last blocks. Each block has kernel size options $\{3, 5, 7\}$ and expansion ratio choices $\{3, 4, 6\}$. Within each block, there are 3 convolution layers. Each convolution layer can independently choose an activation bitwidth and a weight bitwidth. Thus, each block has $(3 \times 3 \times 3^{3 \times 2}) = 6561$ configurations, each stage has $6561^2 \times (1 + 6561 \times (1 + 6561)) \approx 1.85 \times 10^{15}$ configurations. Without considering the elastic input resolution choices, our complete search space contains over $(1.85 \times 10^{15})^5 \approx 2.19 \times 10^{76}$ different quantized architectures.

**Supernet Training and Elastic Quantization**. Following the common practice of starting QAT from a trained full precision network [1], we start by pretraining the supernet without quantizers on ImageNet [39]. We follow the training protocol introduced in [22] to train the supernet. Then, we perform QAT starting from the full precision supernet. Unlike [22, 23] that adopts a progressive shrinking strategy which gradually opens up smaller subnet choices as training progresses, we directly allow all possible quantized subnet options. Our elastic quantization training consists of two stages. In the first stage, we train the supernet for 65 epochs with bitwidth choices $\{2, 3, 4, 32\}$, where a bitwidth of 32 means floating-point precision. Empirically, we found that mixing in the well-trained floating-point layers helps the low precision layers learn and leads to more stable training. Then, we proceed to the second

Table 3: The accuracy of the biggest fixed-precision model after single-stage and two-stage elastic quantization. Single-stage means that we directly train supernet with bitwidth choices $\{2, 3, 4\}$ for 190 epochs, and two-stage means that we first train supernet with bitwidth choices $\{2, 3, 4, 32\}$ for 65 epochs, then continue training the supernet with bitwidth choices $\{2, 3, 4\}$ for 125 epochs. W/A denotes the bitwidth for weights and activation respectively.

| W/A | 4/4 | 3/3 | 2/2 |
|---|---|---|---|
| Single Stage | 74.8% | 73.7% | 66.3% |
| Two Stage | 75.6% | 74.3% | 68.4% |

stage and train the supernet for 125 epochs with only low bitwidth choices $\{2, 3, 4\}$. Table 3 shows that our two-staged training strategy outperforms training a supernet with bitwidth choices $\{2, 3, 4\}$ for the same number of epochs. Our elastic quantization training requires only 190 epochs of training. In comparison, the staged training of fixed precision supernet at each bitwidth in [23] takes 495 epochs in total.

**Subnet Sampling**. Similar to [22, 23, 16], for each supernet training update, we sample and accumulate gradients from multiple subnets. Specifically, we adopt the sandwich rule proposed in [16] by sampling 4 quantized subnets accompanied by the smallest architecture and the largest architecture within the supernet with the entire architecture set to a random bitwidth.

**Multi-objective Evolutionary Search** To balance multiple competing objectives, we leverage multi-objective evolutionary search to produce desirable subnets. Unlike the aging evolution in [22] that only produce one output architecture under a given set of constraints at a time, we adopt the NSGA-II algorithm [40], which outputs a Pareto population at once. As a result, we can quickly access the Pareto front of a trained supernet. At each iteration, the NSGA-II algorithm selects the best-fit population with non-dominated sorting. Then, a crowding distance is calculated to ensure the selected individuals cover the Pareto front evenly without concentrating at a single place. To speed up the search process, we follow [22] and train an accuracy predictor that predicts the accuracy of a given quantized architecture configuration. We use one-hot encoding to encode quantized architecture configurations into a binary vector.

# 5 Experimental Analysis and Results

**Experiment Settings and Implementation Details** We base our codebase on the open-source implementation of [22] under the MIT License, and we follow the exact training procedure on

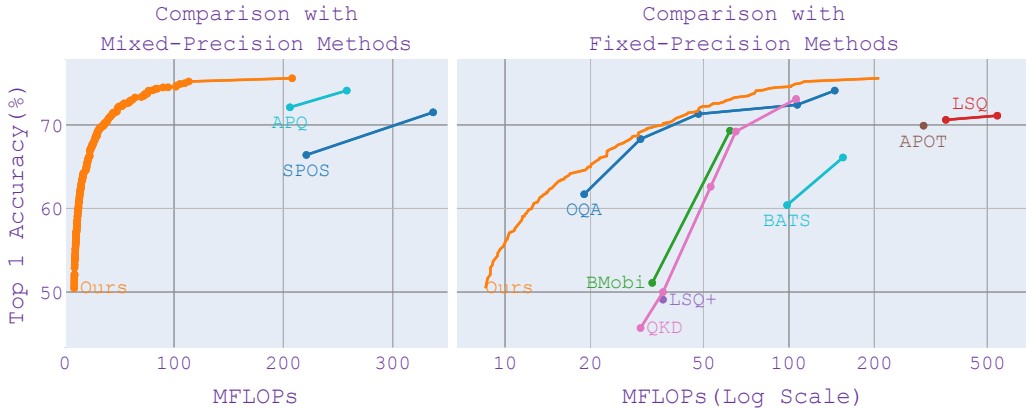

Figure 2: Comparison with state-of-the-art quantization methods on the ImageNet dataset. The left subplot compares our Pareto architectures with those of existing mixed-precision quantized architecture search methods (APQ, SPOS). The right subplot compares our Pareto architectures with existing fixed-precision quantization methods (OQA, BATS, BMobi, LSQ, LSQ+, APOT, QKD). For the ease of viewing, we denote each discovered architecture with a marker only on the left subplot.

ImageNet [39] to obtain the full precision supernet. For both stages of the elastic quantization procedure, we follow the common hyperparameter choice of [22] and use an initial learning rate of 0.08. For all experiments, we clip the global norm of the gradient at 500. We train with a batch size of 2048 across 32 V100 GPUs on our internal cluster. Unless otherwise mentioned, we keep all other settings the same as [22]. After training is complete, we randomly sample 16k quantized subnets and evaluate on 10k validation images sampled from the training set to train the accuracy predictor. During the evolutionary search, we keep a population size of 500 for 1000 generations. For each generation, once we identify the Pareto population based on nondominated sorting and crowding distance, we breed new genotypes through crossover and mutation with a crossover probability of 0.007 and a mutation probability of 0.02. We determine both crossover and mutation probability through grid search. To demonstrate the effectiveness of QFA, we conduct experiments that cover the accuracy and complexity trade-off of searched subnets. For a fair comparison with previous methods, we adopt FLOPs for full precision layers and BitOPs for quantized layers as our complexity measure. While there is no direct conversion from BitOPs to FLOPs, we follow the convention in [21, 23], where given a full precision layer with FLOPs $a$, its quantized counterpart with $m$ bit weight and $n$ bit activation will have a BitOPs of $mn \times a$ and a FLOPs of $(mn \times a)/64$. To be concise, we report our complexity as FLOPs of FP layers + BitOPs of quantized layers / 64 = Total FLOPs.

**Stable Mixed-precision Supernet Training with BQ** To test the effectiveness of BQ in stabilizing supernet training, we attempt to train a baseline supernet with LSQ quantized weights and LSQ+ quantized activations. To test the robustness of low bitwidth training, we only allow bitwidth choices $\{2, 3, 4\}$. We follow the practice of LSQ+ and initialize the scale and offset parameter layer-by-layer through an optimization procedure that minimizes the MSE between quantized output and full precision output at the given layer. However, the training loss diverged at the start of the second learning rate warm-up epoch. Replacing LSQ+ activation quantizer with BQ, we obviate the optimization-based initialization procedure, and the divergent training loss problem no longer occurs.

**Effect of Residual Terms $\gamma$ and $\beta$** To test the effect of $\gamma$ and $\beta$ in BQ, we removed $\gamma$ and $\beta$ from all BQ operations when training the supernet. The training loss diverged within a few update steps.

**Comparing Extreme Value Estimator Choices** To investigate the stability of different extreme value estimators, we train BQ supernet with bitwidth choices $\{2, 3, 4\}$ and plug-in Equations 7, 8, 9 as extreme value estimator. Training loss for both Equations 7 and 9 diverged after the first few updates, and only Equation 8 led to stable training.

**Comparison with SOTA Mixed-Precision Quantized Architecture Search** We compare with existing quantization-aware NAS methods including APQ [20] and SPOS [12], as shown in the left subplot of Figure 2. Our results outperform both APQ and SPOS by a large margin. Thus, in terms of existing mixed-precision quantized architecture search methods, we are able to achieve the best accuracy v.s. FLOPs trade-off.

Table 4: Design cost comparison with state-of-the-art quantized architecture search methods. Our method eliminates the marginal search time. Thus the marginal $CO_2$ emission (lbs) [41] is negligible for search in a new scenario. Here marginal cost means the cost for searching in a new deployment scenario, we use $N$ to denote the number of up-coming deployment scenarios.

| Method | Design cost (GPU hours) | $CO_2e$ (lbs) (marginal) |
|---|---|---|
| SPOS | 288 + 24N | 6.81 |
| APQ | 2400 + 0.5N | 0.14 |
| OQA | 2400 + 0.5N | 0.14 |
| Ours | 1805 + 0.N | **0.** |

**Comparison with SOTA fixed-precision quantized models** We further compare with several strong fixed-precision quantization methods including OQA [23], BATS [21], BMobi [42], LSQ [36], LSQ+ [4], APOT [43], and QKD[44], as shown in Figure 2. Our results demonstrate the competitive performance of quantized subnets discovered by QFA against quantized models produced by state-of-the-art quantization and quantization-aware NAS methods. For models > 20 MFLOPs, we are able to achieve comparable performance as OQA with much less training cost—the OQA supernets require a total of 495 epochs to train, while our mixed-precision supernet took only 190 epochs to train. We achieve SOTA ImageNet top-1 accuracy on models under 20 MFLOPs. Our advantage over OQA at < 20 MFLOPs could be mainly due to the flexibility of our search space to smoothly interpolate between the Pareto frontiers of fixed-precision architectures by mixing layers with varying precision.

**Design Cost Comparison with Quantized Architecture Search Methods** As shown in Table 4, our approach demonstrates better efficiency in terms of GPU hours and marginal carbon emission. We report the total GPU hours for our approach, including 1200 GPU hours of full-precision supernet training, 565 GPU hours of mixed-precision supernet training, and 40 hours of accuracy predictor data collection and training. Running NSGA-II with the trained accuracy predictor does not make use of GPU and finishes within 1 hour. For a fair comparison, we added the GPU hours to train the full precision supernet to the quantized architecture search time reported by the authors of OQA.

## 6 Discussion

While we demonstrate the superior performance of BatchQuant in combination with mixed-precision weight-sharing supernet, there remain many other directions for future investigation. QFA shares some limitations common to many NAS methods with weight-sharing supernets. Since we only visit a small fraction of subnets during training, some subnets may receive insufficient training. Employing techniques [45, 46] targeting this limitation may prove helpful for our especially complex search space. Finally, a thorough theoretical explanation of the effectiveness of BatchQuant on the training dynamics of weight-sharing mixed-precision supernet is still an open problem, and we leave the investigation for future exploration.

## 7 Conclusion

In this paper, we present Quantized-for-All (QFA), a novel mixed-precision quantized architecture search method that can jointly search for architecture and mixed-precision quantization policy combination and deploy quantized subnets at SOTA efficiency without retraining or finetuning. We provide analysis on the challenge of activation quantization in mixed-precision supernet. To allow stable training of QFA supernet, we proposed BatchQuant (BQ), a general, plug-and-play quantizer formulation that stabilizes the mixed-precision supernet training substantially. We demonstrate the robust performance of QFA in combination with BQ under ultra-low bitwidth settings (2/3/4). By leveraging NSGA-II, we produce a family of Pareto architectures at once. Our discovered model family achieves competitive accuracy and computational complexity trade-off in comparison to existing state-of-the-art quantization methods.

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
