# OpenReview forum: "BatchQuant: Quantized-for-all Architecture Search with Robust Quantizer"
_NeurIPS.cc/2021/Conference — NeurIPS 2021 Poster_

### Official Review · Reviewer_JFEL · 2021-07-14

**Rating:** 6
**Confidence:** 3

**Summary:**

This paper proposes a combination of neural architecture search and per-batch quantization statistics calibration in order to quantize neural networks to low bit-width using mixed precision. The proposed NAS method is applied to ImageNet and pareto-optimal networks in the accuracy vs complexity space are found.

**Ethical Concerns:**

Not applicable.

**Limitations And Societal Impact:**

Yes, there is a nice discussion section on that on Page 9.

**Main Review:**

Overall an interesting idea and a generally well written paper. Some questions/comments that I have:

I wish to point out that in eq. (4) $\frac{1-u^N}{1-u} = \sum_{k=0}^{N-1}u^k$ for $u\in[0,1]$. Therefore, we get $E[M_N] = \int_0^1 \frac{1}{\lambda} \sum_{k=0}^{N-1}u^k du = \frac{1}{\lambda} \sum_{k=0}^{N-1}\int_0^1u^k du = \frac{1}{\lambda}\left(1+ \sum_{k=1}^{N-1}\frac{1}{k}\right)$. The conclusion that follows eq. (4) doesn't change much but the correction should be made nonetheless. I also checked reference [38] and did not find a justification for modeling activations as being Laplacian distributed. Can we get some comments on the validity of this assumption. Otherwise, section 3.3 is a good one?

Regarding BatchQuant itself (section 3.4), I am having a hard time understanding how it differs from standard QAT approaches that observe the max value in the instantaneous tensor (such as DoReFaNet). This seems to be what BatchQuant is doing based on the description in lines 202-203. In BatchNorm, there is a moving window that averages mean and variance of the batches temporally and then uses the converged values post-training for inference. That doesn't seem to be what is going on here. Instead, it appears that simple per-tensor scaling is employed. Can we please get some clarification?

Regarding the use of NAS, is there a specific contribution by which mixed-precision is made possible? The introduction and abstract seem to suggest so, however, the description in Section 4 indicates that this is done simply by using prior arts. Either of the two description needs to be corrected for exactness and consistency.

Post-rebuttal: I thank the reviewers for their responses. I am willing to keep my score of a weak accept. But I wish to ask that the authors double check all the math (and crucially its exposition) as well as the details of BatchNorm vs BatchQuant. Ideally, there should be no room for a doubt.

**Time Spent Reviewing:**

2-3 hrs

---

> ### Author Response · Authors · 2021-08-10
> **Response to Reviewer 5**
>
> We thank the reviewer for carefully reviewing our paper, we appreciate the in-depth comments to help us further improve the manuscript.
>
> Q.1. Correctness of Equation (4)
>
> A.1. We thank you for working through our derivation. We checked carefully and found that our derivation agrees with the reviewer's derivation up until the last equality, where $\int_0^1 u^k du$ should yield $\frac{1}{k+1}$ instead of $\frac{1}{k}$.
>
> Q.2. Validity of Laplacian distributed assumption and soundness of Section 3.3
>
> A.2. We thank you for carefully checking the assumption. Modeling the activation distribution for deep networks is usually difficult, as no model is perfect, we agree that we need to be extremely careful to only draw conclusions from where the model is faithful. Similar to [38], the Laplacian distribution is not necessarily the perfect model for activation distribution, but nonetheless a feasible choice to gain intuition quickly about an otherwise complex problem.
>
> In our case, the generalized conclusion we want to draw in Section 3.3 is that training instability arises from the fluctuation of $E[M_n]$ as $n$ varies, which writes formally as $E[M_j] > E[M_i] \quad \forall j > i$. For this condition to hold, we only need $\int_0^\infty (1 - F(t)^j) dt > \int_0^\infty (1 - F(t)^i) dt$ to hold for the activation distribution, which will be valid for any distribution that is not a one-point distribution with CDF $F$ being a shifted unit step function. Furthermore, $E[M_n]$ will continue to grow with $n$ for unbounded distributions. Thus, as long as we can model the activation distribution produced by unbounded activation functions such as ReLU and HSwish as one-side unbounded distributions, the observation of Section 3.3 that $E[M_n]$ will fluctuate as $n$ varies will hold. Note, even though the input distribution of images is bounded, the unbounded activation distribution model quickly becomes applicable as we account for the depth of the network and the training dynamics which push the bound far away.
>
> We also empirically verified that $E[M_n]$ grows in $\mathcal{O}(\log n)$ rate at each layer as we increase $n$. This could potentially mean that the Laplacian distributed assumption faithfully captures the tail behavior of actual activation distribution to some degree.
>
> Q.3. Relation with DoReFa
>
> A.3. Typically quantization requires a scale parameter to scale the activation values to the representable integer range given a bitwidth. Methods such as DoReFa limit the networks to bounded activation functions (HardTanh, ReLU6, etc.) so that the range of activation value is already given. As a result, these methods can work without estimating activation range from the instantaneous tensor. Other methods including BatchQuant do not make the assumption of bounded activation and are thus more generally applicable to pretrained networks with unbounded activation functions. However, to quantize unbounded activation we usually need to estimate a truncated activation range for optimal performance. To our knowledge, there are limited works that directly make use of instantaneous tensor statistics for such estimation. EMA quantizer observes min/max values from instantaneous tensors but smooths the values with an exponential moving average across multiple batches before use. Other methods such as LSQ, LSQ+, PACT learn a parameterized truncation range and only look at tensor value distribution at initialization. BatchQuant combines the best of both worlds by leveraging stabilized range statistics (Equation (8)) within the instantaneous tensor and learning a scale and shift parameter to facilitate learning of a better-truncated activation range.
>
> Q.4. Comparison with BatchNorm.
>
> A.4. Based on [this faithful PyTorch implementation](https://github.com/ptrblck/pytorch_misc/blob/master/batch_norm_manual.py) of BatchNorm, BatchNorm actually leverages the instantaneous batch mean and batch variance during training. The running mean and variance are tracked but are used during inference instead of during training. This makes BatchQuant a close analogy to BatchNorm. We did not mention tracking running statistics explicitly because it is not possible to track the statistics of each individual subnets within the weight-sharing supernet during training. Instead, we perform post-training calibration for both BatchQuant and BatchNorm after selecting a specific subnet. We describe the process at L236-239. Implementation-wise, the code for handling BatchQuant within our submitted supplementary codebase is directly repurposed from the code snippet of BatchNorm by changing the accumulated running statistics.
>
> Q.5. Any contribution that made mixed-precision possible on the NAS search space?
>
> A.5. We thank the reviewer for pointing out the ambiguity. We would like to clarify that, by making mixed-precision possible on the NAS search space, we meant making the supernet training possible, which is more challenging than adding quantization capacity to a search space. By definition of the quantization operation, adding quantization capacity to most existing NAS search space is straightforward. We simply pass the weight and incoming activation of each layer through an added quantizer.
>
> However, as we observed with existing quantizers, training the modified supernet becomes very challenging if not impossible. BatchQuant essentially overcomes such difficulty and allows us to create a compact mixed-precision supernet for the first time and therefore making joint NAS and mixed-precision quantization possible.
>
> We have updated our manuscript to reflect that 1) while inserting quantizers into a NAS search space is not difficult, it showcases the simplicity to plug-and-play BatchQuant into any new search spaces, and 2) having a supernet definition is not sufficient for joint NAS and Quantization to work, we need to overcome the added difficulty in training the expanded supernet, and BatchQuant allows us to do so with ease.

---

### Official Review · Reviewer_552g · 2021-07-15

**Rating:** 4
**Confidence:** 4

**Summary:**

The paper presents the use of batch statics to help stabilize the NAS+Quantization process. It is based on the one-shot NAS idea, but applies the batch idea to assist the quantization policy search during the Neural Architecture Search. The results are compared to some alternative methods and show a faster convergence.

**Limitations And Societal Impact:**

The reviewer didn't find discussions of the limitations of the work or societal impact in the paper. The limitations would be good to add, while the societal impact isn't that important to have for the nature of the work.

**Main Review:**

Strengths:
* The problem tackled in the paper, NAS+Quantization combination, is challenging.
* The idea seems to help speed up the convergence.

Weaknesses:
* Incremental novelty.
* No theoretical analysis on the effects of the introduced technique.
* The evaluation is limited.

Solving the two problems (NAS+Quantization) in a combined manner is challenging. The area has however been explored by prior studies. The proposed idea of using batch statistics to help stabilize the process appears to be a small change rather than anything fundamental. The paper gives no theoretical analysis of the effects of the technique. So it is hard to tell whether the technique is always going to help. A strong comprehensive evaluation could have mitigate the concern. But the evaluation in the paper is quite lightweight, leaving the practical impact of the work unclear.

**Time Spent Reviewing:**

1.5 hours

---

> ### Author Response · Authors · 2021-08-10
> **Response to Reviewer 4**
>
> We thank the reviewer for the constructive review. We hope to provide a few clarifications to address the reviewer’s concerns.
>
> Q.1. Incremental novelty. “Solving the two problems (NAS+Quantization) in a combined manner is challenging. The area has however been explored by prior studies.”
>
> A.1. We thank the reviewer for acknowledging the challenging aspect of combining NAS and Quantization, which is acknowledged by other reviewers as well. Whilst we agree there have been explorations in the field of mixed-precision quantization-aware NAS, we have covered the detailed limitations of previous approaches in both Section 2.2 L93-99 and Table (1) and identified and filled the gaps in the prior explorations. QFA provides novel contributions over the prior studies by being the first to train a compact quantization-aware mixed-precision one-shot supernet, which is the key to faster deployment of Pareto optimal quantized networks as showcased by the substantial performance (Figure (2)) and convergence speed (Table (4)) improvement of our experimental results over those from previous methods. Based on this review, it seems that we may have caused confusion in both Section 2.2 L93-99 and Table (1). We will adjust the wording to better highlight the key differences between QFA and previous approaches.
>
> Q.2. “ The proposed idea of using batch statistics to help stabilize the process appears to be a small change rather than anything fundamental.”
>
> A.2. While we fully agree with the reviewer on the simplicity of BatchQuant, we would like to argue that such simplicity does not come for free, which is instead an added benefit from the important intuition we provided on quantization-aware training of weight-sharing supernet in Section 3.3.
>
> Our discussion in Section 3.3 (and the follow-up discussion with R5 Q2, which we will reflect in the manuscript) sheds light on an important aspect of how unbounded activation distributions will behave during quantization-aware supernet training. Without the insights from Section 3.3, it is not entirely obvious that we should attribute unstable supernet training to the $\Delta$ estimate and what technique we could leverage to stabilize supernet training. The characterization of the training dynamics in section 3.3 made it possible for us to narrow down the root cause and propose a solution accordingly. Thus, we would like to argue that the novelty of our work is not just BatchQuant itself, but more so in providing insights on the training dynamics of mixed-precision weight-sharing supernet. We hope the straightforwardness of BatchQuant in retrospect serves as evidence that our insight is accurate and actionable enough to inspire simple and practical technical advancement in the field.
>
> We thank the reviewer for challenging us to polish the presentation of our contributions. We noticed that we failed to adequately address the value of the insights from Section 3.3 properly in our listed contributions, and we made edits to highlight how the insights enabled us to arrive at BatchQuant as an effective solution.
>
> Q.3. No theoretical analysis on the effect of BatchQuant.
>
> A.3. As we discussed in Section 6, the scope of this paper is focused on providing a practical solution to the problem of compact mixed-precision supernet training. A rigorous analysis of BatchQuant requires combining tools and insights from many active deep learning theory research directions including optimization, quantization-aware training, and weight-sharing neural architecture search. Thus we believe the theory behind the effectiveness of BatchQuant and joint NAS and mixed-precision quantization merits a separate exploration of its own. Nevertheless, we believe our discussion in Section 3.3 as well as the response to R5 Q.2. about the Laplacian distributed assumption shed light on an important aspect of how unbounded activation distributions will behave during quantization-aware training and could potentially serve as a starting point for future research in both practical techniques and involved theoretical analysis.
>
> Q.4. Sufficiency of empirical evaluations.
>
> A.4. We thank the reviewer for commenting on our empirical evaluations. We would like to respond to the reviewer's concern about the sufficiency of our evaluation in supporting the effectiveness of BatchQuant. In terms of dataset, we selected ImageNet as it is the most challenging and well-tested benchmark within both NAS and quantization domain, and models pretrained on ImageNet tend to have better generalizability than those trained on smaller datasets such as CIFAR100. On this front, our evaluation protocol is consistent with many previous works such as [20, 21, 22, 27] published at major venues (CVPR, ECCV, ICLR, NEURIPS). In Section 5, we carefully laid out the ablation studies and showcased the necessity for each component within BatchQuant. The completeness of our argument was acknowledged by other reviewer, that we “established a convincing loop from observation to techniques and results”.
> We understand the reviewer’s concern is partly about the broad applicability of BatchQuant. In addition to existing evaluation, we added results in response to R2 and R3 showing that 1) our searched architectures perform well when finetuned/retrained, and 2) BatchQuant can work as a standalone quantizer for normal quantization-aware training without the weight-sharing supernet.
>
> Q.4. Lack of discussion of limitation
>
> A.4. We have discussed the limitation of our work in Section 6 in hope of outlining the scope of our work and proposing directions for future explorations as acknowledged by other reviewers.

---

### Official Review · Reviewer_Q9hH · 2021-07-16

**Rating:** 7
**Confidence:** 2

**Summary:**

This paper proposed to conduct quantization-aware NAS: in the meanwhile of neural architecture search (basically it is searching subnet within a supernet), it searches for bitwidth allocation for different layer. To stablize the training for quantized activation, it proposed BatchQuant, following similar method of batch normalization.

**Ethics Review Area:**

["I don’t know"]

**Limitations And Societal Impact:**

N.A.

**Main Review:**

1. The protocol of NAS follows previous work, which is less novel. The proposed BatchQuant is interesting: Basically, quantization-aware NAS is adding quantization bit search into NAS. However, BatchQuant is clamined to be necessary for quantization-aware NAS.
2. Can BatchQuant be applied on normal network quantization training ? If possible, are there any experiments?
3. MobileNetV3 is used as spernet, following the convention used in previous work. Is other network possible for supernet?
4. In Line 297-298, accuracy predictor is trained using quantized subnets, how are subnets processed to be inputs for predictor?

**Time Spent Reviewing:**

3

---

> ### Author Response · Authors · 2021-08-10
> **Response to Reviewer 3**
>
> We thank the reviewer for the positive review.
>
> Q.1. Can BatchQuant be applied to normal network quantization training?
>
> A.1. To address this question, we take 4 of our discovered architectures configurations and perform quantization-aware training from floating-point weights using BatchQuant for 125 epochs. Their performance is as shown below.
>
> | QFA Model | A | B | C | D |
> |----------------------------------------|:------:|:------:|:------:|:-------:|
> | Model Complexity (MFLOPs + MBitOPs/64) | 13.97 | 22.88 | 41.8 | 113.39 |
> | Top 1 Accuracy after QFA | 61.92 | 66.86 | 71.02 | 75.19 |
> | Top 1 Accuracy of Standalone BatchQuant | 64.92 | 68.13 | 72.72 | 75.67 |
>
> The results demonstrate that if we know the target quantization policy upfront, we can use BatchQuant as a standalone quantizer choice to provide performance on par with our discovered Pareto Optimal QFA model family.
>
> Q.2. Are other networks possible for supernet?
>
> A.2. BatchQuant is generally applicable and agnostic to specific NAS methods and backbones. We performed initial experiments on ResNet50 backbone. However, ResNet50 is a less compact backbone that cannot achieve competitive Accuracy vs. FLOPs trade-off even when quantized to 2-bit precision, so we decided to switch to MobileNetV3 for better Accuracy vs. FLOPs trade-off. We are eager to test BatchQuant out on other compact network backbones such as EfficientNets in conjunction with other tasks such as semantic segmentation as a part of a future work.
>
> Q.3. How are subnets processed to be inputs for the accuracy predictor?
>
> A.3. We thank the reviewer for bringing this up. Specifically, we convert each hyperparameter choice into a one-hot encoding vector and concatenate all the one-hot vectors into a single input vector to the accuracy predictor. For more details please refer to our codebase submitted within the supplementary material, specifically the ```encode_onehot()``` function within ```nsgaii_search.py```.

---

### Official Review · Reviewer_JJnk · 2021-07-17

**Rating:** 7
**Confidence:** 4

**Summary:**

Authors propose a robust quantizer formulation BatchQuant, which utilizes the batch statistics to provide a more stable quantizer for different quantization levels. With benefit of BatchQuant, they train a compact, single-shot, mix-precision, weight-sharing supernet for NAS. They evaluate BatchQuant on ImageNet.

**Limitations And Societal Impact:**

Yes, they have mentioned and discussed the limitation in Sec 6

**Main Review:**

The paper is well written,

In neural architecture search, it is very challenging to include the mixed precision quantization as a part of policy, since it enlarges the search spaces and increases the quantization instability during the training. BatchQuant helps to alleviate the  instability introduced by quantization and enables to train a large supernet considering different quantization levels of each layer.

BatchQuant estimates the max and min of activations using the current batch statistics and also learns two scalars gamma and beta for quantization. The method is solid and simple.

In experiment, they evaluate on ImageNet and compare with SPOS, APQ, OQA methods. BatchQuant achieves better performance in terms of FLOPS and Top 1 acc.

For rebuttal,
1. Is there any other dataset or other task to evaluate the method on? It is good that when BatchQuant evaluates on other datasets as well
2. Authors choose [2, 3, 4, 32] as bitwidth choices. Can authors elaborate the reason for choosing these bitwidths? It seems there is a huge gap between 32 bits and 4 bits.
3. Even the authors claim that the re-training is not required. But if we re-train the networks, will the performance still get improved?



**Time Spent Reviewing:**

4hr

---

> ### Author Response · Authors · 2021-08-10
> **Response to Reviewer 2**
>
> We thank the reviewer for acknowledging the simplicity of our method and its effectiveness in solving the challenge of training mixed-precision supernet. We would like to address your concerns below and make adjustments to our manuscript accordingly.
>
> Q.1. Try BatchQuant on other datasets or tasks.
>
> A.1. We thank the reviewer for the suggestion. In terms of datasets, we believe the generalizability of our QFA framework is directly inherited from our full precision NAS supernet, which is demonstrated in [45] on multiple datasets. While we are eager to see the performance of QFA on other tasks such as object detection and semantic segmentation, testing QFA in its entirety requires involved investigation on supernet structure and knowledge distillation techniques, which merits a standalone future investigation.
>
> Q.2. Why choose {2, 3, 4, 32} as bitwidth choices.
>
> A.2. We choose bitwidths {2, 3, 4} mainly for a fair comparison with previous works [12, 23]. As for the 32 bitwidth option, as we described in Section 4 of our paper, we only sample it during the first stage of Elastic Quantization to facilitate the learning of other layers that sampled {2, 3, 4} bitwidth choices. During the second stage, we completely remove all the 32 bitwidth options so that we only choose from {2, 3, 4} bitwidth choices. After supernet training is complete, we only consider {2, 3, 4} bitwidth choices for each layer.
>
> Q.3. Will retraining improve the performance of the resulting subnets?
>
> A.3. Following the evaluation protocol in [22, 23], we selected 4 architecture configurations along the Pareto front with increasing complexity and performed finetuning for 25 epochs. The results are as follows, where “Accuracy after QFA” denotes the performance we reported in Figure (2) of our manuscript, and “Finetune@25” means finetuning the selected architectures for an additional 25 epochs after QFA training.
>
> | QFA Model | A | B | C | D |
> |----------------------------------------|:------:|:------:|:------:|:-------:|
> | Model Complexity (MFLOPs + MBitOPs/64) | 13.97 | 22.88 | 41.8 | 113.39 |
> | Top 1 Accuracy after QFA | 61.92 | 66.86 | 71.02 | 75.19 |
> | Top 1 Accuracy after Finetune@25 | 65.15 | 69.54 | 73.37 | 76.48 |
>
> As expected from previous observations in [22, 23], finetuning provides a consistent improvement in model performance. This observation aligns with the limitation we mentioned in Sec 6, as subnets are sampled during training and may compete for performance, a better sampling strategy could potentially improve the final Pareto population. However, such room for improvement does not take away the fact that QFA is able to achieve SOTA Pareto optimal results over previous works without any finetuning, which can efficiently handle a multitude of deployment scenarios at a time.

---

> ### Comment · Reviewer_JJnk · 2021-08-22
> **Good response**
>
> Thanks for the authors' responses. The authors address my concern about the bit-width choice and finetune performance in the rebuttal. But it is still better to report results on other datasets or tasks in the final manuscript since it can further demonstrate the robustness of the method in different applications. I think this paper is good to get accepted and I still keep my original score.

---

### Official Review · Reviewer_HzjN · 2021-07-26

**Rating:** 6
**Confidence:** 4

**Summary:**

This paper summarizes a technique to allow quantizing a weight-sharing superset to with flexible quantization policies to allow choosing different network architecture and quantization policies for desired efficiency-accuracy trade-off. The key observation of this method is that quantizers needs to estimate the maximum and minimum values of a activations. However, the extreme values of activations are dependent on the activation tensor's size. During training, the activation size changes as we sample different architectures, so the extreme value do not converge when using EMA estimator, and the quantizer become unstable during training.  To fix this, this paper proposes a simple method to estimate the extreme values (therefore quantizer parameters) based on current batch statistics. This leads to better stability and performance.

**Ethical Concerns:**

None.

**Limitations And Societal Impact:**

No need for discussion.

**Main Review:**

Strength:
+ This paper proposes a method that effectively combines superset-based architecture search with mixed precision quantization in one-shot, allowing more flexible network configuration for different accuracy-efficiency trade-off.
+ This paper is established a convincing loop from observation (input size difference -> unstable quantizer parameters) to techniques (use batch statistics) and results (improved performance). The story is very clear.

Weaknesses:
- The mathematical derivation in Equation (3, 4) needs improving. E.g., what is the "z" variable in Equation (3)? It is not clearly defined. Also, according to Equation (4), since E(M_n) ~ log (n), seems that the expectation will saturates if n (number of elements in activation) is large enough? Does this imply that if we only sample layers with large activations, we can somehow bypass this problem?
- Any explanation why Equation (8) is the only working estimator?
- MFLOPs calculation questionable. In L308, this paper states that the FLOP operation reported by this paper is computed by mn/a/64. I understand that it is convenient to use one single metric to evaluate the efficiency, but this should not be called as FLOP, which exactly mean floating-point operations.
- Configurations of the optimal networks? It will be valuable to share the configurations of Pareto-optimal network configurations, which is not available in this paper.

Overall this is a neat paper.

**Time Spent Reviewing:**

1.5

---

> ### Author Response · Authors · 2021-08-10
> **Response to Reviewer 1**
>
> We thank the reviewer for taking the time to review our paper, and we appreciate the positive feedback on the clarity of our writing and our chain of reasoning from observation to techniques and results. We hope to address your concerns below and we will make adjustments to our manuscript as well.
>
> Q.1. The notation $z$ in Equation (3) is unclear.
>
> A.1. Thank you for pointing this out. We apologize for the confusion caused by the notation. The notation should be $F(t)^N$ instead, which derives naturally from the previous steps and shows that the CDF of maximum extreme value is the nth power of the CDF of $x$. We have updated both Equation (3) and (4) in our manuscript to reflect the correction.
>
> Q.2. $E[M_n] \sim \log(n)$ will saturate if $n$ is large. If we only sample layers with large activations, can we bypass the problem?
>
> A.2. We appreciate the observation. In theory, the growth of $E[M_n]$ slows down as we increase $n$. However, we found it not immediately relevant in our framework. As we encounter different activation sizes $n$ and $n'$, the quantization-aware training stability hinges on the fluctuation $|E[M_n] - E[M_{n’}]|$. In our search space, activation sizes are scaled multiplicatively, which means $n’ = n / \alpha$ and $|E[M_n] - E[M_{n’}]| \propto \alpha$. Thus larger layers will not necessarily provide better stability than smaller layers. Furthermore, we do not want to overlook the smaller layer options as we want our Pareto architectures to contain good-performing models that are as compact as possible.
>
> Q.3. Why Equation (8) is the only working estimator.
>
> A.3. Based on our reasoning in Section 3.3, we want a $\Delta$ estimator that can properly adapt to the change in activation. We included Equation (7) and (9) with inspiration from EMA and LSQ, respectively. While Equation (7) always exactly captures the activation range without truncating any activation value, its instability comes from its large variance from high sensitivity to outliers, leading to large quantization error. On the other hand, Equation (9) is the least responsive to outliers, however, such inflexibility does not respond well to the higher-order characteristics of the activation distribution (skew, kurtosis) and leads to large quantization error as well. In contrast, Equation (8) enjoys the adaptability of (7) in each channel but robustifies its estimation by taking the average across channels.
>
> Q.4. MFLOPs calculation is questionable.
>
> A.4. Thank you for pointing this out and acknowledging that such calculation is helpful for a fair comparison with previous works [3, 21, 23, 43]. We will update the naming to distinguish the metric from FLOPs.
>
> Q.5. Releasing the configuration of optimal networks.
>
> A.5. We provide the configuration for all 181 of the Pareto optimal networks within the ```pareto_archs.npy``` file and scripts to read the architectures in ```qfa_eval.py``` of our uploaded supplementary material.

---

### Public Comment · ~Shubham_Negi1 · 2022-06-05
**Code Link**

The GitHub link for the code given above doesn't seem to work. Can the authors please update the link?

Thanks

---

### Decision · Program_Chairs · 2021-09-27

**Decision:**

Accept (Poster)

**Comment:**

This paper combines the idea of mixed precision quantization and single shot NAS method, and proposed a single-shot weight sharing quantization method that requires no training during search. 3 out of 4 reviewers acknowledges that the idea is solid and simple. Concerns remain about the novelty. This paper is recommended for acceptance given the effectiveness and satisfying result.